# Advanced Drug Delivery Systems for Renal Disorders

**DOI:** 10.3390/gels9020115

**Published:** 2023-02-01

**Authors:** Batoul Alallam, Hazem Choukaife, Salma Seyam, Vuanghao Lim, Mulham Alfatama

**Affiliations:** 1Advanced Medical and Dental Institute, Universiti Sains Malaysia, Bertam, Kepala Batas 13200, Penang, Malaysia; 2Faculty of Pharmacy, Universiti Sultan Zainal Abidin, Besut Campus, Besut 22200, Terengganu, Malaysia

**Keywords:** renal delivery, kidney disorders, targeted delivery, drug release, glomeruli, nanoparticles, hydrogel

## Abstract

Kidney disease management and treatment are currently causing a substantial global burden. The kidneys are the most important organs in the human urinary system, selectively filtering blood and metabolic waste into urine via the renal glomerulus. Based on charge and/or molecule size, the glomerular filtration apparatus acts as a barrier to therapeutic substances. Therefore, drug distribution to the kidneys is challenging, resulting in therapy failure in a variety of renal illnesses. Hence, different approaches to improve drug delivery across the glomerulus filtration barrier are being investigated. Nanotechnology in medicine has the potential to have a significant impact on human health, from illness prevention to diagnosis and treatment. Nanomaterials with various physicochemical properties, including size, charge, surface and shape, with unique biological attributes, such as low cytotoxicity, high cellular internalization and controllable biodistribution and pharmacokinetics, have demonstrated promising potential in renal therapy. Different types of nanosystems have been employed to deliver drugs to the kidneys. This review highlights the features of the nanomaterials, including the nanoparticles and corresponding hydrogels, in overcoming various barriers of drug delivery to the kidneys. The most common delivery sites and strategies of kidney-targeted drug delivery systems are also discussed.

## 1. Introduction

There has been merging drug development to manage various-related lesions associated with renal diseases. The prevalence of chronic kidney disease (CKD) has been growing significantly for the past two decades in elderly patients, with an estimated 31 million CKD patients in the United States, and it accounts for millions of death globally [1,2]. This emphasizes the significance of evolving of novel drugs and strategies to facilitate renal drug delivery, where therapeutics targeting specific cell types are being investigated. Previous research employed endogenous low-molecular weight proteins, such as immunoglobulin light chain, insulin and lysozymes, in passive targeting, via the natures of their accumulation in the kidneys [3]. Proteins of low-molecular weight are able to diffuse into therapeutics to improve their renal bioaccumulation with a tendency to undergo filtration and deposition in the tubular cells, thus limiting the treatment scope to specific diseases involving proximal tubular epithelial cells [4].

The renal system is a vital system in the mammalian body, serving a variety of tasks. Kidney illnesses are a worldwide health issue that impacts millions of individuals yearly [5]. These issues are generally categorized into CKD [6,7,8] and acute kidney injury (AKI) [9,10,11]. Common kidney disorders include nephrotic syndrome [12], diabetic nephropathy [13], glomerulonephritis [14] and tubulointerstitial fibrosis [15]. The natural progression of these disorders may lead to renal failure, which imposes a significant financial burden on sufferers [16]. In the past decade, high attention was directed to kidney disease. However, the traditional treatment still manifests numerous limitations [17,18]. Current renal therapies possess poor effectiveness by only acting on delaying the advancement of kidney complications, while adverse effects and toxicities limit their long-term application. Thus, developing drug delivery systems targeting the kidneys is critical to improve renal therapeutic effects, while reducing dosages and avoiding damaging adverse effects.

Recently, the application of, and emerged development of, nanomedicines are projected to have a significant influence on drug design and delivery [19]. One of the most important applications of nanotechnology is the administration of anti-cancer drugs, some of which have been authorized for cancer therapy, while others are currently in clinical trials [20,21]. Various types of nanocarriers have been exploited for their therapeutic purposes, such as polymer nanoparticles (NPs) [22,23], inorganic nanoparticles [24], lipid-based nanoparticles [25] and hydrogels [26,27,28]. Nanocarriers are able to carry different therapeutic substances including peptides, ribonucleic acids and small molecule drugs to attain controlled or sustained release in the body [29]. Furthermore, the physicochemical features of nanoparticles can significantly alter their function, allowing prolonged circulation, improved biodistribution and pharmacokinetics, and conferring targeted drug delivery to specific organs, tissues or cells [30]. Many studies reported on the ability of glomerular nanoparticle deposition to endow renal drug delivery for the treatment of kidney diseases [31,32]. Moreover, hydrogels for kidney drug delivery have proven to be effective in achieving local and controlled release of model therapeutics and in protecting and enhancing the effects of concurrently transplanted pro-regenerative cells [33,34,35].

In this paper, we reviewed the application of renal targeting drug delivery systems in several kidney diseases in four sections. Firstly, we briefly epitomize the main anatomy and functions, the pathophysiology of the kidney and the limitations of conventional renal drugs. Secondly, we introduce the main barriers for drug delivery to the kidney. Next, we highlight some delivery sites for kidney-targeted drug delivery systems. Finally, we review some important strategies of renal drug delivery systems.

## 2. Kidney Anatomy and Function

The kidneys help to maintain bodily equilibrium through controlling the blood chemical composition, eliminating excess water, excreting waste items, and synthesizing vital hormones that contribute to the regulation of blood pressure, maintain healthy bones, and impede anemia [36]. The kidney is divided into three sections: the cortex on the outside, the medulla on the inside, and the renal pelvis, which collects and drains urine into the ureter (Figure 1). The nephron is the smallest functional unit in the renal system and accounts for filtration of blood and subsequent reabsorption. Each nephron has a glomerulus surrounded by a Bowman capsule, a proximal tubule, a Henle loop, and a distal tubule connected to a collecting duct. The renal pelvis contains the collecting duct system, and is the place for all nephron drainage. The glomerulus comprises a capillaries network surrounded by podocytes (visceral epithelial cells). Extracellular matrix (ECM) proteins, like collagen IV and laminin that form a basement membrane, along with negatively charged glycoproteins (heparin sulphate proteoglycans) separate the glomerulus endothelial cells from the podocytes. Plasma content below 36 Å Stokes radius is easily filtered in the glomerulus, whereas charged compounds, that require transiting the negatively charged filtration barrier of the glomerulus, are difficult to filter. A daily amount of 180 L is filtered by the glomerulus, that contains a variety of uremic waste products and essential compounds, such as amino acids and glucose that undergo tubular system reabsorption. Furthermore, in order to minimize water loss, the renal tubules reabsorb about 99% of the water that flows through the glomerulus. Due to size, charge constraints or linkage to plasma albumins, the proximal tubular epithelial cells build up uremic waste products that are not filtered in the glomerulus.

## 3. Renal Pathophysiology

Kidneys receive a substantially higher proportion of cardiac output than other critical organs, such as the heart, brain and liver. As a result, the renal tissue is constantly prone to potentially dangerous circulating substances, and the glomerular pressure exposes the capillaries to vascular injury. Immune system diseases, ischemia, drug/xenobiotic toxicity, and genetics are the leading causes of kidney injury [37]. Renal damage causes nephron loss, and, therefore, increases the workload on other functional nephrons. This high workload causes glomerular hypertension and increases nephron size to compensate for the renal function by maintaining glomerular filtration rate (GFR) and intraglomerular pressure through activating the renin–angiotensin system and growth factors (transforming growth factor (TGF) and epidermal growth factor receptor (EGFR)). Protein leakage from glomeruli is caused by podocyte hypertrophy, separation of foot processes of podocytes and rupture of the negatively charged barrier on the glomerular basement membrane (GBM) (Figure 2). Moreover, hyperfiltration causes an increase in the rate of reabsorption by the proximal convoluted tubules (PCTs), which can contribute to tubular injury through lysosomal damage. These events promote inflammatory changes in the tubules, which can lead to interstitial fibrosis and tubular atrophy [38].

CKD impairs drug-metabolizing enzymes and transporter systems, which impacts drug and other substance clearance in the kidneys. CKD patients are reported to possess lower renal CYP27B1 activity as a result of a decrease in 25-hydroxyvitamin D hydroxylation. Non-renal distribution of substances, processed mostly by the liver, is also hampered by renal illness [39]. Acute renal injury is commonly caused by ischemia and toxins, interstitial or glomerular nephritis that is attributed to autoimmune diseases or infection and vascular problems like hypertension. Chronic kidney damage, characterized by tubular atrophy, malfunctioning glomeruli, and interstitial fibrosis, can progress to total renal failure [39]. Renal failure damages the organic anion transporters (OATs) and organic cation transporters (OCTs) that rely on renal secretion of organic ions. The renal excretion of pharmaceuticals and endogenous substances is directly affected by blocking of OATs competitively via different uremic poisons. Renal disease affects not just clearance but also the uptake of medications and organic anions by the liver via transporters.

The majority of renal disorders are caused by other conditions, such as diabetes, hypertension and hypercholesterolemia. Therefore, the medications prescribed to manage these disorders are also utilized to treat or prevent kidney disease. The most often used medications for renal diseases are antibiotics [40], anti-inflammatory agents [41], diuretics, antihypertensive agents [42], anti-hyperglycemic agents [43], fat-lowering agents and hormones [43]. All these treatments exhibit many adverse effects [41,44]. Improved targetability and bioavailability of the drug lower the risk of side effects, which are still the principal areas of concern in the treatment of renal illnesses.

## 4. Kidney Drug Delivery Barriers

The delivery goal of all drugs is to enhance their efficacy by transporting and releasing the drug (passively or actively) to the intended target site in the body while minimizing off-target accumulation. After drug administration, the drug first circulates throughout the bloodstream and accumulates in the target site and once it reaches a certain level, it exerts its therapeutic effect [45]. After drug administration, and depending on the route of drug administration, the drug molecules face some barriers that hamper their deployment to their sites of action. First, some orally administered drugs are largely inactivated before they reach the site of action during the first-pass effect. The principal organs of drug metabolism are the liver and the small intestine in which drugs are chemically modified to facilitate excretion by increasing water solubility [46]. This mainly decreases the pharmacological activity of the drug and/or its half-life. Second, the therapeutics face intravascular enzymes, like proteases and nucleases in circulation that can degrade the active ingredients [47]. Moreover, therapeutic molecules of sizes less than 6 nm can be removed via renal filtration, and, thus, they have extra short renal exposure times which may be insufficient to achieve the therapeutic effect [48]. Next, therapeutic agents that reach the kidney in adequate amounts still cause extrarenal adverse effects. Moreover, the intrarenal transport of a drug may not be optimal concerning the target cell within the kidney, leading to reduced therapeutic efficiency. In addition, the normal renal distribution of therapeutic substances can be impaired in some pathological conditions, like glomerular filtration and/or tubular secretion abnormalities or proteinuria [49]. Therefore, kidney-specific drug targeting could be an appealing approach for overcoming such issues and improving the drug therapeutic index. Moreover, cell-specific drug targeting inside the kidney could provide a valuable pharmacological tool for elucidating the mechanisms of drug action in the kidney.

Developing targeted techniques for renal-specific drug delivery has the potential to reduce undesired side effects and, hence, to improve therapeutic effectiveness. Nanoparticles were developed due to their great potential to target therapeutics to specific tissues [50]; nevertheless, during the delivery of nanoparticles, some barriers hamper their deployment to the cell and impede their targeting capabilities. Blood circulation is a critical factor affecting the efficiency of the majority of systemically administered nanoparticles [45]. Upon systematic administration of nanoparticles, they interact with the extracellular matrix fluids [51], and serum proteins can opsonize on their surfaces, based on the nanoparticles’ physicochemical properties [52]. Opsonization of nanoparticles increases their hydrodynamic diameter considerably, compared to their in vitro diameter, making them more prone to be uptake by the reticuloendothelial system [53]. Besides this, the endothelial barrier limits nanoparticles from passing into the target tissue. Blood enters glomerular capillaries which serve as the filtration barrier for significant quantities of fluid removed from the circulation (Figure 3) [54]. Endothelial fenestrae-lined glomerulus capillaries vascular lumens have intracellular pores (60–80 nm) [55], and are filled with podocytes with a slit diaphragm of 8 nm integral to the filtration. This endothelial barrier between the kidney and blood is very limited in size and charge (Figure 3). After successful extravasation, nanoparticles need to penetrate through the extracellular matrix to reach their target tissues. Small nanoparticles can easily pass, whereas their bigger counterparts face difficulties. Endocytosis varies according to the type of cell. Regardless of the endocytosis type, the rate and extent of cellular uptake are largely governed by the nanoparticles’ physiochemical properties [56]. Thus, NPs’ physicochemical characteristics, dictate their ultimate glomerular accumulation sites.

## 5. Delivery Sites of Renal Drug Delivery Systems

The glomerulus, which is the main unit responsible for blood filtration, is composed of a network of microcapillary tufts and mesangia that support the structure of the tufts. The glomeruli filtrate the blood through the glomerular filtration barrier (GFB) which consists of fenestrated glomerular endothelial cells (GECs), GBM, and podocytes [57]. Glomerulonephritis, which is inflammation of glomeruli, has many forms depending on the cellular location of the injury [58]. Targeted therapy to the injured tissue can not only increase the drug efficacy, but also decrease the severe possible side effects [44]. Targeting different glomerular cells mainly depends on the size exclusion properties of each barrier (Figure 4). Glomerulus-targeted drug delivery systems, sorted according to the targeted site, are presented in Table 1.

### 5.1. Glomerular Endothelial Cells

The endothelium layer bears transcellular windows called fenestrae which are approximately 70–100 nm in diameter, allowing water and small-size molecules to pass through, while inhibiting the movement of objects of bigger sizes. In addition to filtering based on size, the fenestrations are filled with an endothelial glycocalyx which plays a vital role in the filtration of the circulating plasma components, depending on their carried charge [59]. Endothelial glycocalyx is a negatively charged, carbohydrate-rich gel-like structure composed of membrane-bound proteins, “proteoglycans”, such as syndecans and glypicans, cell surface receptors, “glycoproteins”, such as selectins and integrins, and long, linear polysaccharides characterized by their strong negative charge, “glycosaminoglycans (GAGs) such as heparan sulfate, chondroitin sulfate and hyaluronic acid [60]. GECs act as a molecular sieve impeding the passage of blood cells, platelets and negatively-charged molecules, such as albumin [61]. Injury of, or damage to, the GECs can lead to common forms of glomerulonephritis, such as hemolytic uremic syndrome, vasculitis, infection-associated glomerulonephritis and membranoproliferative glomerulonephritis [31]. Diabetes mellitus-induced proteinuria also correlates to damage of GECs [62]. Asgeirsdottir et al. developed a GECs-targeted delivery system composed of corticosteroid dexamethasone-loaded liposomes conjugated with anti-E-selectin antibodies (AbEsel-Liposomes). E-statin is a receptor expressed on activated endothelial cells to recruit different kinds of immune cells during inflammation [63]. The results demonstrated that AbEsel-Liposomes were accumulated in the kidney 3.6 times higher than the non-targeted liposomes and successfully decreased the glomerular endothelial activation and albuminuria after seven days [64]. This study demonstrated the future applicability of using disease-specific epitopes on the surface of GECs for targeted drug delivery. Such an approach has particular potential in anti-glomerulonephritis therapies of potent anti-inflammatory drugs that, till today, cannot be used, due to severe systemic adverse effects.

### 5.2. Glomerular Basement Membrane

The GBM is a central, non-cellular trilaminar membrane composed of lamina rara interna next to the endothelial cells, lamina densa in the middle, and the outermost lamina rara externa next to the podocytes [65]. GMB is an extracellular matrix consisting of 144 distinct secreted proteins from podocytes and GECs [66]. The most abundant ones in normal GBM are collagen subtypes (α3, α4, and α5), laminin (α5, β2, and γ1), nidogen and heparin sulfate proteoglycan (agrin and perlecan) [67]. A network of fibrils is formed from these proteins with meshes of diameter 4–10 nm [62]. The GMB has a high negative charge due to the presence of heparin sulphate, the main component of the anionic proteoglycan. GMB with this strong negative charge was thought to be the ultimate barrier for proteinuria by forming electrostatic repulsion with other anionic proteins, such as albumin, that prevents their filtration [68]. However, when negatively-charged and neutral Ficoll filtration on isolated GBM was investigated, all the particles passed through the basement membrane equally [69]. In another study, a reduction of about fivefold anionic sites in the GBM of transgenic mice by heparanase negated proteinuria [70].

### 5.3. Podocytes

Podocytes, which are also called visceral epithelial cells, are highly specialized epithelial cells that form the main filtration barrier in the glomerulus. Podocytes have unique structural features of interdigitating foot processes with slit diaphragms of around 25 nm width anchored in between the adjacent foot processes [71]. Podocytes are attached to GBM by cell–cell adherence (ex: vinculin, E-cadherin, and nephrin), and a specific matrix of proteins (ex: VCAM-1, tenascin-C) [72]. Podocytes have limited ability in repair or regeneration, and, thus, in the case of podocyte injury, the podocytes detach from the glomerulus and are cleared into the urine, which leads to reduced podocyte numbers [73]. Damage to podocytes or loss of podocytes is a characteristic feature of many nephrotic syndromes, such as minimal-change disease, lupus nephritis, membranous nephropathy, and focal segmental glomerulosclerosis, which are all accompanied with proteinuria [74]. Podocytes are well known not only for being a filtration barrier, but also for maintaining the normal structure of the glomerulus by secreting angiopoietin 1 and vascular endothelial growth factor 1 (VEGF-1) [75]. Glomerular podocytes also highly express the neonatal Fc receptor, which binds to IgG and prevents its degradation [76]. Fc receptor is also the binding site for albumin [77]. Accordingly, Wu et al., successfully prepared 10 nm albumin methylprednisolone nanoparticles targeting the Fc receptor on glomerular podocytes. The uptake of the formulated nanoparticles in the FcRn-expressing human podocytes was 36-fold higher compared to the uptake in the non-FcRn-expressing control cells [74]. In a different study, Visweswaran et al. designed a novel formulation of anti-VCAM-1 antibody-modified rapamycin nanoparticles to target VCAM-1 receptor on TNFα-activated podocytes. Compared to free rapamycin and non-targeted rapamycin nanoparticles, anti-VCAM-1-rapamycin nanoparticles inhibited the migration of podocytes more effectively, indicating the potential of targeting podocytes through VCAM-1 targeting nanoparticles [58]. However, the most significant challenge for podocyte-targeting nanoparticles is to pass through GFB to reach podocytes in vivo as all the above examples were conducted in in vitro cell models.

### 5.4. Mesangial Cells

Mesangial cells are irregularly shaped cells that conserve the structural integrity of the glomerular tufts [61]. Mesangial cells also account for secreting and maintaining the mesangial matrix, and producing soluble factors by which communication with other glomerular cells is possible [78]. Mesangial cell injury is implicated in many renal diseases, such as diabetic nephropathy and glomerular sclerosis [79]. Thus, mesangial cell-targeting drugs would be useful in various kidney disease treatments. To deliver drugs to the mesangial cells, the drug only has to pass the fenestration of the GECs (100–700 nm), due to the absence of both GBM and podocytes between the GECs and the mesangial cells [62]. Accordingly, particle size seems to be the key factor and the most important principle in targeting the loaded drug to the mesangial cells, instead of podocytes [77]. Choi et al., successfully prepared gold-based nanoparticles with a size of 75 nm that effectively accumulated in the mesangia, suggesting that effective targeting of mesangial cells could be achieved by formulating nanoparticles with sizes between 70 and 100 nm [80].

**Table 1 gels-09-00115-t001:** Renal delivery system sorted according to the targeted site.

Targeted Site	Target	Delivery System	Loaded Drug	Size (nm)	Disease	Refs.
Glomerular endothelial cells	E-selectin	Liposomes conjugated with anti-E-selectin antibodies	Dexamethas-one	121 ± 20	Glomerulonephritis	[64]
Glomerular basement membrane	-	cyclodextrin-containing polymer-based siRNA nanoparticles	siRNA	60 to 100	Normal	[81]
Mesangial cells	-	siRNA-loaded polycationic cyclodextrin nanoparticles (siRNA/CDPNPs)	siRNA	~70	Normal	[82]
Mesangial cells	-	Celastrol-albumin nanoparticles	Celastrol	95	(Thy1.1) Nephritis	[83]
Mesangial cells	-	PEG- (TRX-20)-modified liposomes	Triptolide	100	Membranous nephropathy	[84]
Podocyte	Neonatal Fc receptor	Bovine serum albumin-methylprednisolone conjugate Nanoparticles (BSA633-MP)	Prednisolone	10	Nephrotic syndrome	[74]
Podocyte	VCAM-1 receptor	Lipid-based nanocarrier SAINT-O-Somes	Rapamycin	128 ± 4	TNFα-activated podocytes (mimic the inflammatory condition)	[58]

## 6. Nanoparticle Factors for Enhanced Renal Accumulation

### 6.1. Nanoparticle Size

Due to glomerular filtration barrier size selectivity, the size of NPs has a major impact on their biodistribution and therapeutic potential [85]. In healthy kidneys, it is believed that the overall cutoff for NPs to pass through the glomerular filtration barrier is about 30–50 kDa or 8–10 nm. Although small NPs may penetrate deeper into tissues, particles exhibiting less than this threshold are able to traverse the glomerular filtration barrier, enter the tubule system, and, consequently, are quickly cleared via renal excretion and phagocytosis [80,86]. Studies evaluated the biodistribution of nanoparticles with sizes ranging from 1 to 10 nm and demonstrated a renal clearance of >40% of the injected dose (ID) 24 h post-injection [86,87,88,89,90,91,92]. For instance, Singh et al., found that silicon nanoparticles, with a size of 2.4 ± 0.5 nm, were ~100% cleared into the urine 24 h post-injection [87]. Low renal retention typically coincides with a high clearance rate of these small NPs, conferring them to be unsuitable as renal drug delivery systems. Table 2 shows that the majority of NPs with a size range from 1 to 10 nm possess <5% kidney accumulation at 24 h, as they probably undergo renal clearance [86,91]. On the other hand, Du et al., claimed that ultrasmall NPs, with a diameter of ~1 nm, interacted with the endothelial glycocalyx of the glomerular filtration barrier, endowing reduced clearance rate and prolonged retention time in the kidney compared to larger NPs (2.5–6 nm) [93]. These results emphasized the significance of the glycocalyx in the interaction of ultrasmall nanoparticles with the kidneys.

On the other hand, NPs with a size range of 10–20 nm are also able to transverse the glomerular filtration barrier [74,94,95,96]. The main strategy to achieve high renal accumulation is by proper selection of the NPs starting material. Soft organic polymer-based materials, such as albumin-based nanoparticles and PEGylated amphiphile micelles [95], can be squeezed into the pores of podocytes. Although some larger NPs (20–100 nm) find it difficult to pass completely through the glomerular filtration barrier, they may enter the kidneys through disassembly into smaller components. Cyclodextrin-containing polymer-based siRNA nanoparticles, with a hydrodynamic size of 60–100 nm (zeta potential of 10.6 mV), remain intact in the bloodstream but disassemble upon contact with the negatively-charged proteoglycans, enabling passage into the kidneys [81]. Nanoparticles composed of ferric iron, tannic acid, and PLG-g-mPEG (FeTNPs) (HD = 75 nm) were also reported to be disassembled by deferoxamine, via the formation of ferrioxamine with iron [97]. Deferoxamine mesylate, the iron chelation agent, was injected into mice 24 h after intravenous FeTNP administration in order to trigger the dynamic disassembling of FeTNPs, and the dissembled FeTNPs, which were smaller than 6 nm in diameter, were found to be excreted into urine by renal clearance [97]. Hence, disassembly of larger particles into smaller components could be a strategy to facilitate nanoparticles entering the kidneys. Apart from disassembly as a strategy to pass into the kidney, these NPs (20–100 nm) can also accumulate in the kidneys as a result of their interaction with the mesangial cells. Choi et al. hypothesized that PEG-gold nanoparticles, with a diameter of ~79 nm, accumulated in the kidney (4.5% ID) as they interacted, particularly with mesangial cells [80]. Outside the glomerulus capillaries, the mesangium and fenestrated endothelium are linked to each other, hence NPs > 15 nm might pass through the fenestrated endothelium but not the glomerular basement membrane (GBM) and might diffuse and aggregate in the mesangium [80]. It is, thus, necessary to investigate mesangium size selectivity to develop therapeutic mesangium-targeted NPs for the treatment of mesangium-related kidney illnesses, like mesangial cell abnormalities and mesangio-proliferative glomerulonephritis [98,99].

Even though NPs with sizes more than 100 nm are too large to pass through the GFB, they can still enter the kidney during tubular secretion and excrete into the urine [97,100,101,102]. Endogenous metabolites and xenobiotics are generally secreted through the tubule epithelial cells into the proximal tubule [101,102,103], and larger NPs > 100 nm may also utilize this mechanism for kidney access. For instance, Wyss et al. [102] reported that PLGA NPs, with diameters of 130–160 nm, were localized in the proximal convoluted tubules and tubule lumen of TEM kidney sections. In addition, PLGA-b-mPEG NPs, with a size of about 400 nm, were localized to the proximal tubule cells and exhibited the highest fluorescence intensity in the kidney among all other organs, as verified by immunofluorescence imaging [103]. PEG-modified magnetite cubes and clusters with hydrodynamic size of 139 nm translocated from the peritubular capillaries into tubule cells through in vivo intravital imaging in mice in [101]. After injection of the clusters, the fluorescence signal of the clusters peaked at approximately 5 min and started to decrease due to the excretion of the nanoparticles into urine [101]. These studies confirmed larger nanoparticles (>100 nm) could also be directed into the kidneys through secretion by tubule epithelial cells from the peritubular capillaries. Specifically, after leaving the glomerulus via the efferent arterial, and entering the peritubular capillaries, larger nanoparticles >100 nm entered the renal tubule system through exocytosis from peritubular capillary epithelial cells and endocytosis into proximal tubular epithelial cells [103].

The distribution of NPs in the kidneys is size-dependent due to many barriers that exist between the kidneys and the surrounding fluids or tissues [85]. The size of NPs impacts the mechanism employed to enter the kidneys. NPs with a size of about ~1 nm are harnessed to accumulate in the glycocalyx, whereas those with a size of ~80–90 nm are leveraged to target the mesangium. NPs with a size >100 nm can enter the kidneys via exocytosis from epithelial cells of the peritubular capillary into the proximal tubular (Figure 3). As a result, the size of NPs can be tailored to target specific cell types in the kidney, giving renal nanomedicine more flexibility and possibilities.

**Table 2 gels-09-00115-t002:** The influence of the physicochemical properties (size and charge) of various NPs on their renal clearance and accumulation.

NPs	Size (nm)	Charge (mV)	Renal Accumulation (% ID) 24 h Post Injection *	Renal Clearance (% ID)	Refs.
[^64^Cu]Cu-1,4,7- triazacyclononane-triacetic acid tagged with near-infrared dye (IR800-CW)- silicon NPs	2.4 ± 0.5			100% ID 24 h post injection	[87]
Glutathione-coated gold NPs (GS-AuNPs) Gold NPs coated with glutathione and cysteamine (GC-AuNPs)	GS-AuNPs = 2.1 ± 0.4 GC-AuNPs = 2.9 ± 0.3			40-50% ID 24 h post injection	[86]
Cysteine-coated gold NPs	3.5 ± 0.9		8.8 ± 2.0	More than 50% ID 24 h post injection	[88]
Luminescent glutathione coated copper NPs (GS-CuNPs)	2.2		0.6	78.5 ± 3.5% ID 24 h post injection	[89]
Glutathione-coated silver NPs (GS-AgNPs) Glutathione-coated Au/Ag NPs (GS-Au/AgNPs) Glutathione-coated gold NPs (GS-Au)	~3.1			GS-AgNPs = 51.36% ID GS-Au/Ag(1)NPs = 52.99% ID GS-Au/Ag(2)NPs = 48.69% ID GS-AuNPs = 45.57% ID 48 h post injection.	[90]
Core-shell silica-based NPs (C dots)	3.3 and 6.0			C dots (3.3 nm) = 73% ID 48 h post injection C dots (6.0 nm) = 64% ID 48 h post injection	[91]
GS-[^198^Au]AuNPs	3.0 ± 0.4			~50% ID 48 h post injection	[92]
Gold NPs	1.4		~1.9		[104]
Quantum dots	5.6		~15% ID/g		[105]
Gold NPs	3.1		~15% ID/g		[106]
Carbon nanotubes	25		0.6% ID/g		[94]
PEGylated kidney-targeting peptide amphiphile micelles and PEGylated amphiphile micelles	Targeted micelles = 15 nm Non-targeted micelles = 12 nm		Targeted micelles = ~35% and non-targeted micelles = 26% of total fluorescence, in the kidneys		[95]
Copper sulfide nanodots	5.6	+2.9		95% ID 24 h post injection	[107]
Silicon NPs	2.4	+5.4		100% ID 24 h post injection	[103]
Gold NPs	2.9	−27		42% ID 24 h post injection	[86]
Gold nanoparticles coated with cysteine (Cys-AuNPs) Gold nanoparticles coated with glycine-cysteine (Gly-Cys-AuNPs)	Cys-AuNPs = 2.69 ± 0.46 nm Gly-Cys-AuNPs = 3.12 ± 0.61 nm	Cys-AuNPs = −12.52 Gly-Cys-AuNPs = −27.33		Cys-AuNPs = 21.5% ID 24 h post injection Gly-Cys-AuNPs = 41.6% ID 24 h post injection	[106]

(*): %ID, used as unit to measure the nanoparticle quantity in each organ, while % ID/g used as the unit for biodistribution results of nanoparticles.

### 6.2. Nanoparticle Surface Charge

The capillary wall of glomerular vessels in humans acts as a charge-selective barrier that prevents charged molecule filtration, which governs the biodistribution of NPs inside the kidney [108]. Although small NPs with size 1–10 nm are small enough to pass through the GFB, the charge selectivity of the GFB was found to affect the interaction of nanoparticles with different charges. It was observed that positively-charged NPs tend to pass the GFB more easily than negatively-charged NPs, due to electrostatic repulsion. For example, the kidney accumulation of cationic polyethylenimine-conjugated quantum dots with zeta potential of +23.4 mV was ~15% ID/g 24 h after injection, which was higher than anionic mercaptosuccinic acid-capped quantum dots with zeta potential of −52 mV, although both particles showed hydrodynamic size less than 6 nm [105]. The main two critical variables that contribute to the impact of surface charge on renal filtration are the possible interactions between NPs and serum proteins; and the interactions between NPs and the capillary wall of the glomerulus [109]. Adsorption of serum proteins on NPs increases the hydrodynamic diameter of NPs, thus decreasing their renal filtration.

In terms of renal clearance, positively-charged NPs are cleared more rapidly compared to negatively-charged ones, due to facilitated passage through the glomerular filtration barrier and the tubule system [103,107]. Electrostatic interactions allow positively-charged NPs (6–8 nm) to cross the renal filtration barrier; however, negative surface charges or neutral counterparts with the same hydrodynamic diameter are more likely to be prevented from passing through the filtering barrier, due to charge repulsion (Table 1) [80]. For example, copper sulfide nanodots with zeta potential of +2.9 mV and silicon NPs with zeta potential of +5.4 mV were 95% ID and 100% ID cleared after 24 h post-injection, respectively [10,103]. In 7contrast, the renal clearance of negatively charged nanoparticles of similar size was found to be lower 24 h after injection. Gold NPs with zeta potential of −27 mV showed renal clearance of 42% ID, and 41.6% ID for gold nanoparticles with zeta potential of −27.33 mV [86,106]. Since positively-charged nanoparticles are generally cleared faster than negatively-charged nanoparticles, the overall charge of nanoparticles should be turned into a negative charge in order to achieve longer retention in the body. Interestingly, some literature claimed that NPs with strong negative charge could pass the glomerular filtration barrier faster than NPs with slightly negative or positive charge, in some conditions [86,110]. Generally, NPs are able to exhibit more retention time in the body when their net charge is negative regardless of the starting material types.

A few techniques have been widely used to manipulate NPs’ surface charges. including conjugation of charged ligands, such as peptides [111], and the coating of NPs’ surfaces [112]. Neutral or zwitterionic coatings on nanoparticle surfaces can reduce the chance of opsonization and avoid capture by the liver reticuloendothelial system (RES) [110,113]. On the contrary, NPs with a high net charge are prone to opsonization, and, hence, they can achieve higher renal accumulation due to the size-dependent glomerular filtration barrier [114]. Zwitterionic coatings, such as zwitterionic peptides, KKEEE, and EEKKK, have been found to facilitate NP entry to the kidney in vivo [110,111]. Thus, this approach could be utilized to aid NPs in passing the glomerular filtration barrier and rapidly accessing the kidneys. Upon guaranteeing the entry of nanocarriers to the kidney, a cell-specific targeting ligand could be incorporated to enhance renal retention. For instance, conjugation antibodies to NPs could facilitate their glomerular localization [115,116]. Moreover, in kidney disorders such as inflammation, which often results in acidosis, conjugating specific targeting ligands, like stimulus-responsive moieties or bioactive-penetrating molecules, to NPs could enhance glomerular localization by acting as smart carriers that respond to their surroundings [115,116].

### 6.3. Nanoparticle Shape

The principle “form follows function” has significantly affected the architecture and characteristics designs of NPs, including rheological dynamics, cellular uptake, and drug loading capacity [117,118,119]. A variety of NPs have been formed with various shapes. Some shapes, such as rods, wires, and fibers, have one long, non-nano scale dimension, while other shapes, such as sheets, disks, and chips, have two lateral dimensions greater than the third [120]. Compared to spherical NPs, oblate shapes survive longer in circulation (Table 3) [121]. When one end of a NP with a high aspect ratio interacts with macrophages, higher fluid shear stresses are on its other free end, and, hence, the NP can be cleared away from the surface of the cell prior to endocytosis and internalization by cells [122]. As a result, the NP is less susceptible to macrophage clearance and stays in circulation longer compared to its spherical analog. Moreover, successful endocytosis is dependent on the curvature of the local membrane caused by the interaction of the NPs with a circulating phagocytic cell [123]. While high aspect ratio NPs are unlikely to cause curvature when their long dimension comes into contact with a cell, spherical NPs can promote curvature at each interaction during circulation [124]. High aspect ratio NPs are capable of evading phagocytic uptake and have the ability to remain in circulation for a longer time interval. Thus, this may be beneficial when tailoring NPs to provide improved kidney accumulation [125]. NPs of high aspect ratio can readily access the kidney by aligning the long axis towards the filtration slit. Since the width of the NP is less than the cutoff threshold, the length can be increased by orders of magnitude without impairing kidney filtration [126].

The effects of the size and surface characteristics of NPs on drug renal delivery have been well studied, whereas studies on the influence of particle shape are scarce. Considering the glomerular endothelial pores size restriction and low permeability of GBM, it was previously believed that possessing a diameter greater than that of these pores could not be utilized. However, the exceptional pharmacokinetic profile of the oblate shaped NPs has made them appealing carriers for renal glomerular clearance [118]. Accordingly, numerous experiments have been conducted in this direction. Novel high aspect ratios are now being investigated for use in a variety of biomedical applications [127,128]. Perhaps these findings may permit the development of new nanomaterials that are capable of specifically targeting podocytes and the epithelial cells of Bowman space.

**Table 3 gels-09-00115-t003:** The influence of the NPs with various shapes on their renal clearance and accumulation.

NPs	Shape	Size (nm)	Advantage over Control Particles	Refs.
Single walled carbon nanotubes	Rod	1.2 × 100−1000	65% ID increase in renal clearance	[126]
Malleable poly(glycidyl methacrylate) (L-PGMA)	Rod	43	4.2% ID/g increase in renal accumulation	[128]
Mesoporous silica NPs	Rod	159	16% renal increase in renal accumulation	[129]
RNA nanosquare	Square	10	66% increase in renal accumulation	[130]
Gold nanostar	Star	55	~40% increase in renal accumulation	[131]

### 6.4. Material Choice of Nanoparticles

NPs made from a variety of materials have been claimed to have different renal accumulations. For example, low material density metal NPs <5 nm, like silica nanoparticles, have low targeting ability and high renal clearance. This tendency is related to the density-dependent circulation rate of ultrasmall metal NPs in blood circulation [90]. NPs of more than 10 g/cm^3^ have a greater buoyancy force to the endothelium and may reach there more quickly compared to NPs with a lower density (<10 g/cm^3^) [132]. As a result, the densest NPs do not remain in the blood vessel central flow lines, where fluid velocity is the highest [90]. On the other hand, NPs with a lower density circulate more rapidly and are dispersed more easily throughout the body, leading to a shorter blood retention period, and, subsequently, increased renal clearance. For instance, silicon nanoparticles with a hydrodynamic diameter of 3.3 nm and low density had almost complete renal clearance (98%) at 24 h injection dose [87], while gold nanoparticles with a hydrodynamic diameter of 2.5 nm and about eight-fold higher density exhibited two-fold lower renal clearance at 24 h after injection [133].

## 7. Strategies of Renal Drug Delivery Systems

### 7.1. Small Molecule Prodrugs

Pharmacologically, prodrugs are inactive derivatives of a parent drug molecule that result from impermanent chemical modifications of bioactive compounds, which are activated at a specific target site to confer a therapeutic effect. Prodrugs should be ideally stable in the blood against enzymatic degradation, but rapidly convert into an active compound at the site of action [134]. The chemical modification is usually intended to reduce drug molecule toxicity, and enhance membrane solubility or permeability [135]. Typically, they consist of a small molecular inactive agent conjugated to a macromolecular carrier as a transporter or a lipophilicity enhancer [136].

#### 7.1.1. Folate-Modified Prodrugs

The kidneys play a crucial role in preventing folate loss. Folate is metabolized in vivo as 5-methylenetetrahydrofolate, that is filtered into the glomerulus before reabsorption at the renal vascular circulation by a folate-binding protein (FBP), localized mostly in the proximal tubular epithelium. Hence, folate can be employed to deliver drugs to proximal tubular cells as a ligand [44]. Mathias et al. successfully produced [^99m^Tc]DTPA-folate and assessed its potential as a Folate Receptor-Targeted Radiopharmaceutical [137]. The [^99m^Tc]DTPA-folate complex biodistribution in the kidney reached about 21% of the injected dose per gram tissue after 4 h of intravenous administration in Athymic Male Mice with KB Xenografts. The biodistribution level of a complex in the kidney was lowered to 2.3% when free folic acid was administered at 1–2 min prior to complex injection. Similarly, Trump et al. found that after 4 h of intravenous injection, the biodistribution of [^99m^Tc](CO)3-DTPA-folate in the kidney was nearly 47% of the injected dose per gram tissue [138].

Moreover, in pioneering work by Wang et al., DTPA–folate conjugate was synthesized by attaching folic acid to a diethylenetriaminepentaacetic acid via an ethylenediamine spacer [139]. The conjugate was excreted rapidly in urine following intravenous administration to a normal rat. A pharmacodynamic study on athymic tumor-bearing mice showed a good uptake of DTPA–folate in tumor tissue with a large distribution in the kidneys after 1 h of intravenous administration [140]. In addition, Shan, et al., produced folic acid-conjugated paclitaxel–Evans blue prodrug for targeted cancer therapy [141]. The targeting ability study was carried out using tumor-bearing mice. Their findings showed that the concentration of paclitaxel in the kidneys was increased from 5769 ng/g to 9181 ng/g after intravenous injection of free-paclitaxel and folic acid-conjugated paclitaxel, respectively. Thus, folate-conjugated small molecules may be an excellent carrier for renal drug targeting and the physicochemical properties of the complex may possess an essential role in tissue targeting.

#### 7.1.2. Sugar-Modified Prodrugs

Recently, the significance of sugar-modified peptides in renal-targeted drug delivery has attracted a lot of attention in pharmacology and pathology. The structure of the sugar plays an essential role in renal selectivity and efficient uptake of the glycosylated peptides through binding to the renal microsomal fraction that enhances peptide distribution in the proximal tubules [142,143].

Suzuki et al. synthesized glycosylated peptide (^[3H]^Glc-S-C8-AVP) to determine the inhibitory glycosides effects on the kidney membrane fraction using different glycosylated derivatives. Their findings indicated that the alkylglucoside structure (Glc-S-C8-) was required for targeting of the kidney. The targeting efficiency was affected by the sugar moieties type, especially the type of linkage, structure of the peptide, and the length of the alkyl chain, as well as the charge and size of the molecule [143]. On the other hand, Shirota et al. characterized the renal targeting efficiency and limitations of alkylglucoside carriers [144]. The researchers combined alkylglucoside vector with acylated poly-L-lysine at different molecular weights to study the effect of the size of derivatized ligands with alkylglucoside. The study of tissue distribution found that molecular weight had a negative effect on the accumulation of alkylglucoside-acylated poly-L-lysine accumulation in the kidneys of mice. In addition, alkylglucoside was combined with anionic, neutral, or cationic tyrosine derivatives to investigate the charge effects on the specific binding to kidney membranes. According to these results, the anionic charge of the alkylglucoside-tyrosine conjugate led to reduction in the renal targeting efficiency of the conjugate.

Moreover, Lin and co-workers prepared a prednisolone succinate-glucosamine conjugate (PSG) and a 2-deoxy-2-aminodiglucose-prednisolone conjugate (DPC) as a potential targeted delivery system of prednisolone in the kidneys [145]. The cytotoxicity and cellular uptake studies using HK-2 and MDCK cell lines showed that the conjugate PSG decreased the cytotoxicity, as well as improved the cellular uptake to 2.2 times higher, compared with prednisolone. DPC-enhanced kidney-specific localization in vivo, in which the drug concentration was 4.9-fold greater than that of prednisolone, was observed The researchers concluded that 2-glucosamine, as well as 2-deoxy-2-aminodiglucose, could be potential carriers for kidney-targeting drug delivery.

In addition, Liang, Zhen, et al. conjugated zidovudine with chitosan oligomers to increase the half-life and elimination time of zidovudine in human plasma and kidneys [146]. The conjugate was evaluated with reference to in vitro release in mice plasma and renal homogenate after intravenous administration. The results indicated that the mean retention time of the conjugate was 1.5 h, compared with 0.59 h for zidovudine without chitosan oligomers, as well as the conjugate enhancing the accumulation of zidovudine in the kidney higher than in any other organ. Hence, in order to improve the treatment of acute kidney injury through enhanced rapid distribution in the kidney and enhanced retention time in the renal tubule, Lui et al. prepared an l-serine-modified chitosan-based carrier [147]. This system showed rapid accumulation and long-term retention in renal tubules resulting from the native cationic charge of chitosan and the specific interactions between serine and kidney injury molecules.

#### 7.1.3. Amino Acid-Modified Prodrugs

Some endogenous enzymes exist in a high concentration in kidneys, especially those involved in amino acids, such as γ-glutamyltranspeptidase and L-decarboxylation [44,148]. On this basis, it was envisaged that chemical modification of drugs with substrates of these enzymes would increase the concentration of drug molecules in proximal tubular cells as a result of the effect of the relevant enzyme [44]. In this regard, Wilk et al. synthesized γ-glutamyl-dopamine (GGDA) chemically and enzymatically, followed by determination of the tissue distribution of dopamine in the kidney after injection into mice [148]. After equivalent dose administration, the dopamine concentration in kidneys produced by GGDA was 5-fold higher than that of L-dopa. The renal blood flow was significantly improved with an insignificant effect on the blood pressure and on the heart. In addition, by administration of GGDA orally, the results showed that the concentration of free dopamine was high in urine compared to that in the plasma [149,150]. The researchers concluded that GGDA could be considered a potential specific strategy to deliver dopamine to the kidney.

Moreover, in order to improve the renal targeting of the parent drug prednisolone, Su et al. synthesized an N-acetyl-glutamyl prednisone prodrug for evaluating in vivo distribution and the bone mineral densities (BMD) in rats [151]. The obtained findings showed that, compared with the parent prednisone, the renal concentration of prodrug was increased and its effect on bone density was reduced. Renal endothelial cells are a major part of the filtration system in the kidneys. Throughout transplant-associated ischemia, endothelial cells are the main target of complement-activated injury, resulting in delayed posttransplant function [152]. Durigutto et al. generated a targeted delivery system to renal endothelium by coupling a neutralizing anti-C5 antibody with a cyclic RGD peptide [153]. The novel system showed preferential localization to ischemic endothelial cells in a rat for renal ischaemia reperfusion injury (IRI). The injected anti-C5 antibody RGD reduced the renal injury level without a significant effect on circulating levels of C5. The antibody conjugate was suggested as a novel target for drugs to prevent post-transplant IRI as well as in transplant medicine.

### 7.2. Antibody Modified Carriers

Antibodies are unable to be filtered via the glomerulus because of their high molecular weight (about 150 kDa), and, thus, they cannot be applied as a drug delivery system for targeting the renal proximal tubuli. However, radiolabeled end products of antibody fragments showed prolonged renal radioactivity, as well as radiolabeled monoclonal antibody fragments, which have an accumulation in renal tubuli according to the studies of anti-cancer therapy [154,155,156]. Although the prolonged and increased accumulation of fragments in the kidneys are undesirable side effects of these cancer treatment strategies, taking the use of antibody fragments as a renal carrier system into consideration could be beneficial. The advantage of incorporating antibody fragments for therapeutic administration into proximal tubular cells is the targeting of disease-related growth factor receptors, such as EGF receptors and TGF-β receptors, located in the proximal tubular cells on its basolateral and apical membranes [157,158]. Antibody-drug conjugates may, thus, have 2-fold inhibitory effects: first, preventing natural ligands from binding to the receptor, and second, delivering the drug after internalization of the drug-antibody conjugate by target cells.

Li et al. employed an antibody fragment F(ab’)2 in order to target plasmalemma vesicle-associated protein (PV1), an endothelial cell-specific protein localized in caveolae, fenestrations, and trans-endothelial channels as a structural component [159]. By finding the suitable binding affinity of an antibody toward PV1, the developed delivery system showed continual retention in mice kidneys at 24 h, whereas the isotype control F(ab′)2 was eliminated rapidly in the urine with a significant reduction in signaling in the kidney. The researchers suggested that PV1-targeted F(ab′)2 is potentially a useful system for delivering therapeutic agents to the kidneys. On the other hand, Kvirkvelia et al. attempted to treat nephritis by using a human monoclonal antibody (F1.1) as a vehicle for delivering the drug to glomeruli via directly targeting the noncollagenous-1 domain (NC1) of 3(IV) collagen [160]. After coupling PGE2 and dexamethasone to F1.1, the glomerular localization, as well as the conjugation capacity to modify disease, were estimated in mice with established nephritis. The conjugates demonstrated high efficacy and reduced systemic effects and the blood urea nitrogen levels were also improved, compared to the untreated mice.

Podocytes, also known as visceral epithelial cells, are the last barrier of the kidney filtration and their function is critical in proteinuria remission. They are harmed in a variety of diseases with immune and non-immune causes. In order to target the visceral epithelial cells within the glomerulus, an anti-mouse podocyte antibody was modified by means of enzymatic cleavage and by coupling with a protamine molecule [161]. Protamine has a positive surface charge, and, thus, it can bind to nucleic acids that are negatively charged, such as siRNA. Furthermore, protamine provides protection for nucleic acids against degradation. By loading siRNA specific for nephrin into this antibody, the mRNA levels of nephrin in mice were significantly decreased, pointing to the specificity of the delivery system.

### 7.3. Macromolecular Carriers

Macromolecular carriers are considered to be very useful vehicles for targeting therapeutic molecules to the kidney, in which low molecular weight glomerular protein (LMWP) can accumulate in the kidneys selectively. Generally, macromolecular carriers are small molecular weight (MW <30,000 Da), biologically active, proteins in the circulatory system, including peptide hormones (such as insulin), immune proteins (such as light chain immunoglobulin), and enzymes (such as lysozyme) [44]. Ordinarily, the macromolecular carriers have a molecular weight larger than that of the encapsulated drug, and the kinetics of the protein carrier is superior to the intrinsic kinetics of the drug. In particular, the filtration of LMWP occurs through the glomerulus and the reabsorption through the renal tubules. The properties of LMWP, such as non-immunogenic properties, as well as its several functional groups, confer on LMWP the potential to be a promising delivery system for different drugs. Drugs conjugate with LMWP through different methods, such as ester, peptide, amide, and disulfide bonds [44,162]. The macromolecular carrier–drug conjugate is removed from the circulation and then the drug is released under enzymatic or chemical hydrolysis, while the activation occurs in lysosomes. Ideally, the synthesis of the conjugate of drug–LMWP requires a high degree of skill due to the several active groups of LMWP that are extremely vulnerable to self-aggregation [44].

In this regard, Kok et al. studied the impact of the chemical modification of primary amino groups on the pharmacokinetic profile of drug–LMWP conjugates. The organic anion fluorescein isothiocyanate (FITC) was used as a model drug [163]. In order to study the effect of charge modifications on the FITC–LMWP lysozyme (LZM) conjugate distribution, they prepared a series of conjugates with different amounts of primary amino groups. The final prepared conjugates had several determined amounts of primary amino groups of 8.5 (FITC-cat-LZM), 6.5 (FITC-LZM), or 0.2 (FITC-Suc-LZM). The findings showed that the charge of the products had an effect on the biodistribution of FITC–LZM conjugate. The tubular reabsorption was reduced while the excretion of the conjugate was increased into the urine as a result of reducing the amount of primary amino groups. Moreover, the renal reabsorption and the extrarenal distribution of the conjugate with partial loss of renal selectivity were enhanced by increasing the amount of primary amino groups.

LMWP lysozyme is considered a suitable drug system for renal drug targeting. In another study by Kok et al., they synthesized two different FITC–LZM conjugates with and without positive charge to study urinary excretion [164]. The positively charged conjugate was synthesized by reacting fluorescein isothiocyanate (FITC) with lysozyme, whereas the non-charged conjugate was synthesized by reacting succinic anhydride with the remaining free primary amino groups of the FITC–LZM. The obtained data indicated that the urinary excretion of a drug–LMWP conjugate was enhanced via decreasing the positive charge of the carrier surface. This system could be a promising candidate for drug delivery to the bladder.

### 7.4. Water Soluble Polymeric Carriers

Several research projects have shown that water soluble polymers have a beneficial effect on kidney-targeted drug delivery approaches. The accumulation in the renal tubule of any polymer is mainly related to several determinations, such as the final molecular weight, the type of the monomeric unit, the anionic group of the monomer, and the content of the anionic monomer [165].

One example, polyvinylpyrrolidone (PVP), with low molecular weight, is excreted in the urine without an accumulation in the kidneys [166]. However, in the study of Kodaira et al., intravenous administration of carboxylated PVP showed an improved renal accumulation compared to sulfonated PVPs, in which around 30% of the administered dose was observed in renal tubules; mainly the proximal tubular epithelial cells [167].

Due to the safety property, as well as the strong kidney-targeted ability of Poly (vinylpyrrolidone-co-dimethyl maleic acid) (PVD) as a drug carrier, Yamamoto et al. studied the relationship between the molecular weight of PVD and its renal accumulation in mice [168]. The findings showed that the molecular weight of 6–8 kDa led to the highest renal accumulation, in which 80% of the intravenously-administered dose accumulated in the kidneys for 3 h.

Liu et al. applied atom transfer radical polymerization to synthesize a panel of polymers in order to determine the impact of molecular weight as well as anionic charge density on kidney targeting and distribution in mice [169]. The observed results showed that kidney-specific polymer accumulation improved due to the anionic monomer content, but not the molecular weight. The experimental focal segmental glomerulosclerosis enhanced kidney accumulation of anionic polymers; and anionic polymers accumulated mainly in proximal tubule cells, with little distribution in kidney glomeruli.

Kamada et al. synthesized polyvinylpyrrolidone-co-dimethyl maleic anhydride [poly(VP-co-DMMAn)] to evaluate its use as a kidney-targeted drug carrier [166]. The conjugate led to an increase in accumulation and retention in the kidneys, compared to unconjugated drugs, without any toxic effect, in which the accumulation was around 80% for 24 h. with about 40% remaining in the kidneys for 96 h after intravenous administration to mice. In contrast, the distribution of polyvinylpyrrolidone with the same molecular weight showed a random distribution in vivo. In addition, modifying poly(VP-co-DMMAn) with superoxide dismutase led to a high accumulation in the kidneys of mice, conjugated with an accelerated recovery from acute renal failure. unlike the native superoxide and polyvinylpyrrolidone modified superoxide dismutase.

### 7.5. Nanoparticles

The distinctive size of nanoparticles has been widely applied to develop an effective targeted delivery system. Particles and colloids with a size of 5–7 nm can pass the glomerular filtration barrier, and, thus, they may be used in the targeting of the tubular area. Otherwise, systems with a size range of 30–150 nm do not penetrate into primary urine, unless in the case of damage in the glomerular filtration barrier by disease or the if the particles have been degraded into particles <10 nm [55]. The particle size of 80 nm was attributed as being the maximum glomerular accumulation, regardless of some exceptions, such as the distinct chemical composition of each carrier system, as well as extra-small carriers (2 nm) which find it difficult to pass the glycocalyx and are usually removed by the liver [170].

Gao et al. prepared low molecular weight chitosan nanoparticle-loadedsiRNA duplexes by means of ionic gelation as a therapeutic targeting strategy for various kidney diseases [171]. Chitosan/siRNA nanoparticles, with a size range of 75 ± 25 nm, were administered to chimeric mice with conditional knockout of the megalin gene. The results showed a specific delivery after nanoparticles administration to proximal tubule epithelial cells in mice kidneys with a residence time of more than 48 h. The specific uptake of proximal tubule epithelial cells was mediated by megalin, and that led to a reduced expression of the water channel aquaporin 1 (AQP1) by up to 50%. This system might be a potential platform for treating many kidney diseases via targeting of siRNAs, or drugs, to proximal tubule epithelial cells.

Recently, chitosan polymer was used to synthesize nanoparticle-loaded metformin to investigate its efficiency as an oral drug delivery carrier for chronic kidney disease [172]. Chitosan nanoparticles were produced by ionic gelation followed by several experiments to examine their physiochemical and muco-adhesion properties, as well as their release profile, in a low pH environment. Metformin was encapsulated into chitosan nanoparticles in order to modify the paracellular permeation and improve the oral bioavailability of metformin. Upon administration of the nanoparticles (~145 nm diameter) by oral gavage to a murine model of polycystic kidney disease, a higher metformin bioavailability, as well as a lower cyst growth, was observed compared to the free drug. The nitrogen, creatinine, and blood urea levels were found to remain similar to untreated mice, elucidating the absence of nephrotoxicity. This study highlighted chitosan nanoparticles as a potential oral delivery platform for polycystic kidney disease.

PLGA nanoparticles were also applied for renal applications as a therapeutic agent delivery [173,174]. In this approach, Tang et al. successfully employed PLGA nanoparticles in plasmid DNA (pDNA) delivery, in which pDNA was embedded in calcium phosphate (CaPi) and encapsulated into PLGA nanoparticles [175]. The transfection efficiency was enhanced after optimizing pDNA loading efficiency as well as pDNA release kinetics, compared to conventional methods of plasmid delivery (e.g., lipofectamine). In addition, a significant improvement in the transfection efficiency of the applied nanoparticles on human embryonic kidney (HEK 293) cells was observed compared to pDNA-loaded PLGA nanoparticles as well as the CaPi-pDNA embedded PLGA microparticles.

In another study, Williams and his team investigated renal safety and selectivity after the administration of mesoscale nanoparticles that were prepared by conjugating poly (lactic-co-glycolic acid) with polyethylene glycol (PLGA-PEG) [100]. The direct administration of these particles intravenously showed an increase of around 26-fold in renal accumulation, greater than in any other organ. The ex vivo imaging and intravital microscopy exhibited the delivery of mesoscale nanoparticles into proximal tubule cells. In addition, the mice that were treated with the nanoparticles did not show any systemic consequences, liver impairment, immune reaction, or renal impairment. This study portends more developments on the targeted drug delivery of renal tubules. The recent advances in kidney-targeted drug delivery systems using nanotechnology are shown in (Table 4).

### 7.6. Liposomes

Liposomes are used as a drug delivery platform, due to their following features: controllable release profile, stability in vitro and in vivo, targeted drug delivery, and disease site localization. Liposomes have been widely studied for drug delivery and they are used in several current marketed products. Singh et al. prepared unilamellar vesicles (SUVs) containing methotrexate [(MTX) SUVs)] linked to Dal K29 [182]. After 2 h of vesicle incubation with CaKi-1 cancer cells (human kidney), the binding to CaKi-1 cells increased about 8-fold more than unlinked (MTX)SUVs and around 6-fold more than nonspecific mouse myeloma IgGl-linked (MTX)SUV. In addition, the obtained results of the colony inhibition assay showed that the prepared system was, respectively, 40 and 5 times better than free MTX and Dal K29-MTX in inhibiting the growth of CaKi-1 cells.

By using prednisolone phosphate (PSLP) as a model drug, the ability of PEG-modified liposomes containing TRX-20 (TRX-liposomes) to target mesangial cells and their pharmacokinetic behavior was studied by Morimoto and his team, using a rat experimental glomerulonephritis model [183]. The results showed that TRX–liposomes targeted glomeruli in the kidney cortex successfully via binding to chondroitin proteoglycans. In particular, TRX–liposomes showed their effect using a much lower dose than that required of PEG–liposomal formulations or the conventional injection of PSLP. The distinct pharmaceutical properties of TRX–liposomes support the idea that they are a candidate system for targeting inflamed glomerular mesangial cells in glomerulonephritis therapy.

In the study by Tuffin et al., mesangial cells were targeted by the prepared liposomes with Fab′ fragments of OX7 mAb (OX7-IL) [184]. However, a single injection of a low dose of doxorubicin encapsulated in OX7-IL was administrated intravenously and showed extensive glomerular damage without adverse effects on other kidney tissues or other organs.

Li et al. prepared a system in order to deliver celastrol to interstitial myofibroblasts, that was related, specifically, to renal fibrogenesis [185]. The loading of celastrol into linear pentapeptide Cys-Arg-Glu-Lys-Ala (CREKA)-coupled liposomes led to an alleviation in renal fibrosis induced by unilateral ureteral obstruction (UUO) in mice, with much lower toxic effect than the free drug. This study suggested that loading celastrol in CREKA–liposomes could increase its therapeutic effect and decrease its systemic toxicity as a novel strategy in renal fibrosis treatment. In addition, the ability of CREKA-coupled liposomes to bind to fibronectin could permit imaging application to diagnose renal fibrosis. All the delivery system strategies are summarized in Figure 5.

### 7.7. Hydrogel

Hydrogels can be utilized to encapsulate various therapeutics that are useful in the treatment of kidney diseases. Existing kidney-targeted drug delivery hydrogels can be classified as follows: (1) cell-seeded to enhance paracrine activity and cell survival, (2) therapeutic-loaded for controlled release, and (3) hydrogel/NPS composites (such as micelle- and exosome-loaded hydrogels). The majority of hydrogels used for kidney therapy are administered by intrarenal/intracapsular injection. Intracapsular injection of hydrogels has gained popularity in recent developments in the utilization of hydrogels for kidney drug delivery and has proven to be effective in achieving local and controlled release of therapeutics and in protecting, as well as enhancing, the effects of concurrently transplanted pro-regenerative cells. However, the majority of the hydrogel reported on were used to deliver cell therapy and used invasive delivery through injection (Table 5).

Stem cell therapy, including mesenchymal stem cells (MSCs), and endothelial progenitor cells (EPCs) has been demonstrated to treat chronic kidney disease. It involves the use of a paracrine activity to stimulate the repair of damaged kidney tissues [193,194]. Nevertheless, cell therapies are often limited by the poor survival and density of transplanted cells. A number of nanomaterials have been developed to improve the efficiency of stem cell therapy for patients with kidney disease [186]. Generally, these hydrogels are composed of polymeric substances with high biocompatibility and a natural origin. Some of the hydrogels used are collagen/poly γ-glutamic acid-loaded with antioxidant α-lipoic acid [195]; cross-linked HA/collagen loaded with therapeutics such as stromal cell-derived factor-1 [196,197,198]; and decellularized bovine pericardium [199]. In rodent models of adriamycin-induced nephrotoxicity [197], cisplatin-induced renal dysfunction [196], and lipopolysaccharide-induced endotoxemia [196,197,198], hydrogels enhanced the retention of co-transplanted stem cells, as well as the paracrine effect. This helped to protect stem cells from the damaging effects of an inflammatory environment.

Another strategy using hydrogels was employed to deliver therapeutics. Using a “plum-pudding” design, Qin et al. developed an injectable micelle–hydrogel hybrid. The pudding was a hyaluronic acid (HA) hydrogel cross-linked with the plums, Pluronic F127-methacrylate self-assembled micelles. Micelles containing celastrol (anti-inflammatory medication) were introduced into HA hydrogel containing anti-transforming growth factor-1 (TGF-1) antibodies. The targeted and extended (over 21 days) release of each treatment inside the kidneys reduced inflammatory markers and slowed the formation of renal interstitial fibrosis in unilateral ureteral obstruction mice models [34].

NPs are frequently being coupled with hydrogels to produce hybrid nanomaterials for multi-step controlled delivery [200]. These hybrids have been utilized to treat kidney disease, but in an invasive way. Hence, delivering such hybrids in a non-invasive way may serve as a valuable reference for establishing delivery of hydrogels to the renal system. In addition, these hybrids can be used to support the infiltrating cells of patients with damaged or dysfunctional kidney tissues. The most commonly used biocompatible and biodegradable materials include natural polymers, such as gelatin, collagen, alginate, HA, and chitosan. The properties of natural and endogenous polymers should be considered when designing hydrogels. For instance, HA can be found in healthy tissues with low molecular weight form (900 kDa) [201]. High molecular weight (HA) can be found in kidney tissues, which can help mediate the development of kidney diseases. On the other hand, low molecular weight HA can be detrimental to the development of kidney diseases [202]. In acute kidney injury, the induction of interleukin-10 by the blood vessel stimulated high molecular weight HA. It was found that this condition could protect the kidney from damage and fibrosis. It was also observed that high molecular weight HA could serve as a biomarker for the progression of fibrosis following kidney transplantation. High molecular weight HA was shown to maintain the structure of the vascular endothelium and reduce inflammation in patients with IgA and diabetic nephropathy. The results of the study suggested that the use of high molecular weight HA hydrogels and biomimetics could be beneficial for the treatment of kidney disease.

## 8. Conclusions

Some nanomaterials have found their way onto the clinical stage with the rapid development of nanomedicine. The ultimate goal of designing kidney-targeted therapy is to control the concentrations of drugs and improve the therapeutic effect of the kidney while minimizing toxic and other negative side effects of the medicine. The nanodrug delivery system offers great control on particle size, charge, shape, and surface characteristics, conferring specific targeting for varied kidney pathogenesis. This review elucidated the mechanisms and techniques of kidney-targeted nanodrugs with regard to the characteristics and anatomical structure of the kidney to equip theoretical bases for kidney disease therapy. Nanotechnology has provided sound evidence in kidney-targeted delivery to treat various diseases. However, the associated limitations, including the high cost, low yield, possible toxicity to non-target organs, low targeting efficacy, and poor in vivo stability should be controlled. Currently, research on kidney-targeted drug delivery systems is mostly focused on discovering acceptable carriers and enhancing targeting efficiency, while the release rules and metabolic processes of drug delivery systems after they penetrate target cells are still being investigated. Kidney-targeted medication delivery systems will play a significant role in the treatment of kidney disorders as a result of an in-depth study on kidney diseases and the application of nanotechnology.

## Figures and Tables

**Figure 1 gels-09-00115-f001:**
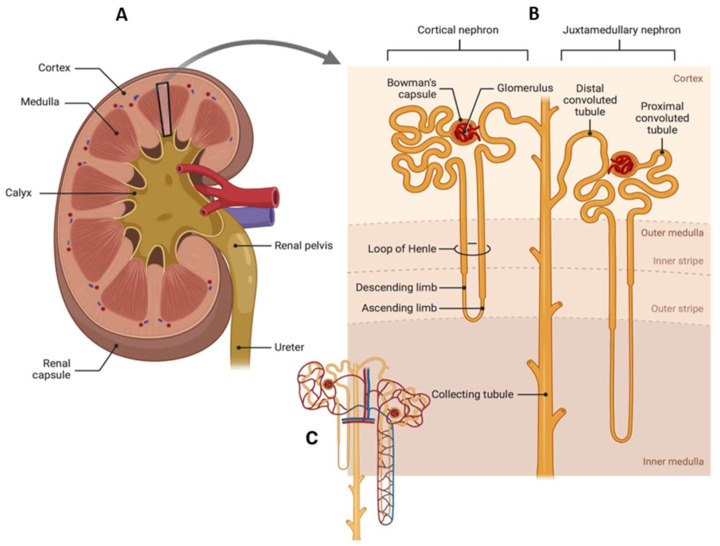
Schematic presentation of the renal system. (**A**) human kidney, (**B**) nephron, and (**C**) nephron with blood capillaries.

**Figure 2 gels-09-00115-f002:**
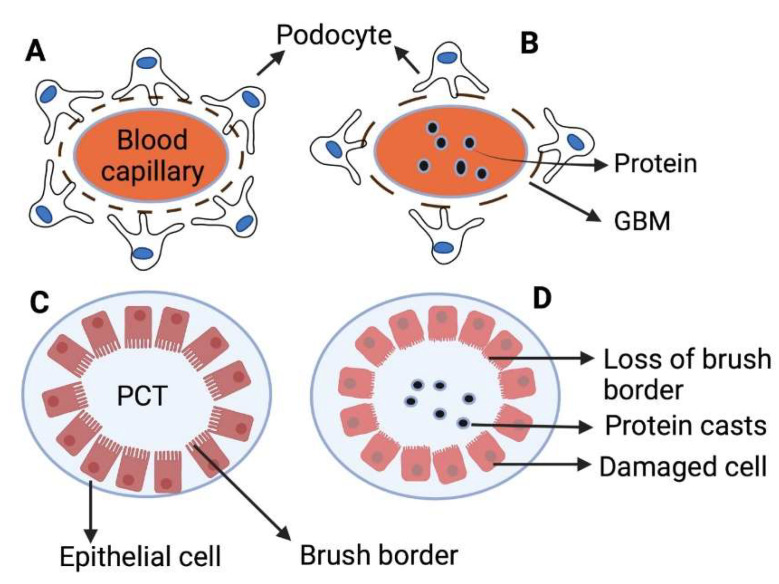
Schematic presentation of podocytes and proximal convoluted tubule. (**A**) healthy podocytes and glomerular basement membrane (GBM), (**B**) swollen proximal convoluted tubule (PCT) in injury condition explicit by the gaps in GBM, (**C**) healthy proximal convoluted tubule with good brush border epithelium, and (**D**) injured proximal convoluted tubule with lack of brush border.

**Figure 3 gels-09-00115-f003:**
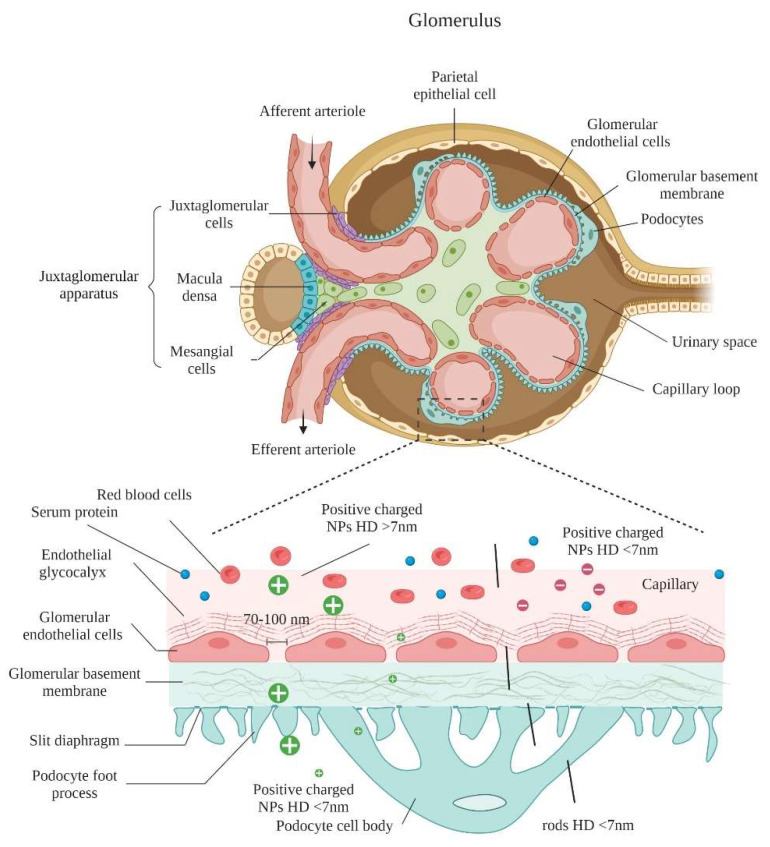
Kidney-targeted drug delivery. The nanoparticles filtration process in the glomerulus as function of their size, charge, and shape. HD indicates hydrodynamic size of the nanoparticles.

**Figure 4 gels-09-00115-f004:**
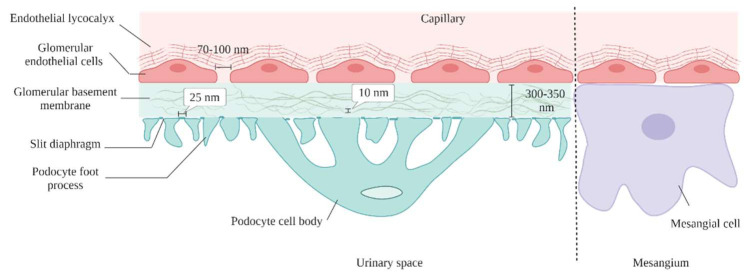
Schematic presentation of the glomerular filtration barriers.

**Figure 5 gels-09-00115-f005:**
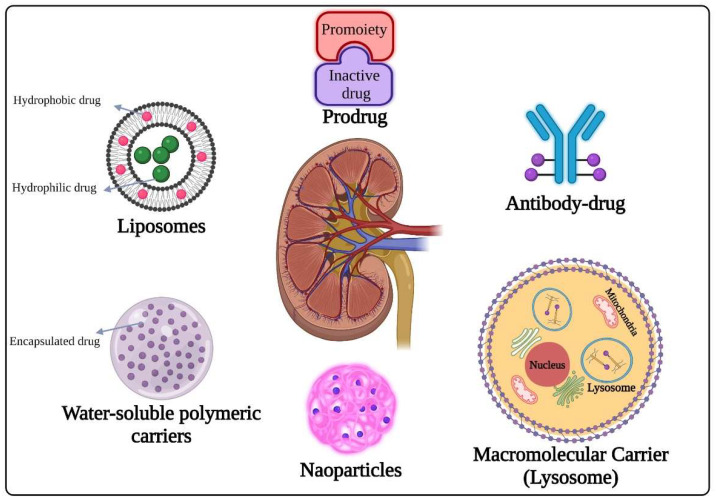
Different strategies of renal drug delivery system.

**Table 4 gels-09-00115-t004:** Nanotechnology applications in the development of kidney-targeted drug delivery systems.

Materials	Size (nm)	Characteristics	Target	Application	Refs.
PLGA	207 ± 5	Calcium phosphate embedded for plasmid(p)DNA delivery	Kidney	Promising vectors for gene delivery	[175]
Chitosan	38–45	Catechol-derived low molecular weight chitosan/Doxorubicin	Kidney	Renal fibrosis	[176]
Liposome	100–150	pDNA-encapsulating/SS-cleavable and pH-activated lipid	Kidney	Renal cell carcinoma	[177]
Gold	-	Nanoparticle arrays as biosensor	Kidney	Chronic kidney disease	[178]
Cationic cyclodextrin	60–100	cationic cyclodextrin-containing polymer (CDP)-based siRNA nanoparticles	Glomerular basement membrane	Nucleic acid delivery	[81]
Functionalized chitosan	2.2–3.6	Functionalized chitosan/quantum dot nano-hybrids	Phosphate metabolites	Treating hyperphosphataemic patients Kidney failure	[179]
low molecular weight chitosan	75 ± 25	Chitosan/siRNA nanoparticles	Proximal tubule epithelial cells (PTECs)	Knockdown of specific genes in (ptecs) Kidney diseases	[171]
Chitosan	150	Metformin-loaded chitosan nanoparticles	(Intestines) improve oral bioavailability of metformin	Polycystic kidney disease Chronic kidney disease	[172]
PLGA-PEG	200	FITC-labelled renal tubular-targeting peptide modified PLGA-PEG nanoparticles	Renal proximal tubules	Chronic kidney disease	[173]
PLGA	207 ± 5	Calcium phosphate-embedded PLGA nanoparticles	Embryonic kidney cells	Gene delivery	[175]
PLGA-PEG	347.6 ± 21.0	Poly(lactic-*co*-glycolic acid) conjugated to polyethylene glycol (PLGA-PEG) nanoparticles	Proximal tubule cells	Targeted drug delivery of renal tubules	[100]
Gold	75 ± 25	PEGylated Gold-based nanoparticles	Mesangium of the kidney	Kidney diseases	[80]
Dextran dendrimer	5	Dextran-based nanoparticles poly(amido amine) dendrimer nanoparticles	Renal tubular epithelial cells	-	[180]
PEG-PLGA	77.8 ± 0.5	Lambda light chains (LCs) attached to PEGylated polylactic-co-glycolic acid (PLGA) nanoparticles	Proximal tubule epithelial cells	Management of non-oncologic/oncologic renal disorders	[181]

**Table 5 gels-09-00115-t005:** Hydrogel-based therapies for kidney-targeted drug delivery systems.

Hydrogel Carrier	Cargo	Route/Target	Refs.
Hyaluronic acid/collagen/polyethylene glycol hydrogel	Mesenchymal stem cells, and endothelial progenitor cells	Intracapsular injection	[186]
Chitosan hydrogel	Mesenchymal stem cells	Intracapsular injection	[98]
Self-assembling peptide hydrogel	Mesenchymal stem cells	Intracapsular injection	[33]
Biotin/chitosan hydrogels	Mesenchymal stem cells-derived extracellular vesicles	Intracapsular injection	[187]
(Arginine-Glycine-Aspartate) peptide hydrogel	Mesenchymal stem cells -derived extracellular vesicles	Intracapsular injection	[188]
Chitosan hydrogel	Nitric oxide-donor enzyme-prodrug system	Intracapsular injection	[189]
Collagen hydrogel	Extracellular vesicles	Intracapsular injection	[190]
Peptide hydrogel	Mitochondria antioxidants Mito-2,2,6,6-tetramethylpiperidine-N-oxyl (TEMPO)	Intracapsular injection/ mitochondria	[191]
PEG hydrogel	Fibroblast growth factor and murine epidermal growth factor	Intracapsular injection	[192]
Collagen hydrogel	Prostaglandin E_2_	Intracapsular injection	[35]
Micelle-hyaluronic acid hydrogel	Celastrol/anti-transforming growth factor-β1 antibody	Intracapsular injection	[34]
Folate-conjugated micelle nanoparticles into polyvinyl alcohol MN patches patch	Rhodamine B	Folate receptor targeting	[28]

## Data Availability

Not applicable.

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
