# Peer review of "Advanced Drug Delivery Systems for Renal Disorders"

_gels, 2023, doi:10.3390/gels9020115_

Round 1

Reviewer 1 Report

Generally, this review manuscript is overall acceptable, though the writing style and format are not smooth and not easy to follow. The original of the topic is not that novel, though acceptable. It seems that the authors expand the introduction/overview parts regarding the renal anatomy and functions and renal pathophysiology/limitations of conventional renal drugs to avoid the similarity to the available published review articles. These introduction/overview parts are too long. Shortening the introduction/overview parts and rearrangement of the heading might improve the manuscript quality and flow.

1. This review manuscript describes a broad overview of the nanomaterials namely nanoparticles and hydrogels as the kidney-targeted drug delivery systems. the content composed of 4 parts: kidney anatomy and functions, renal pathophysiology and limitations of conventional renal drugs, the main barriers to kidney drug delivery, and delivery sites of kidney-targeted drug delivery systems.

However, the overview parts regarding the renal anatomy and functions, renal pathophysiology and limitations of conventional renal drugs, and the main barriers to kidney drug delivery are very long. These overview part can be shortened. The content of the main objective starts from page no. 7, heading no. 6-8 (6. Nanoparticles Factors for Enhanced Renal Accumulation, 7. Delivery Sites of Renal Drug Delivery Systems, 8 Strategies of Renal Drug Delivery Systems).  The arrangement of the main content should be also reordered to be 7, 6, and 8.

The manuscript should be checked carefully to revise writing mistakes. There are a few spelling mistakes (I believe that this mistake will be corrected and checked during preparing the published version by MDPI).  The name of drug delivery systems should always contain the “s”, e.g., Line 21:  nanomaterial, nanoparticle and hydrogel must be corrected as n nanomaterials, nanoparticles and hydrogels.

2. The topic is relevant in the field. However, if considering the original or novelty, this review is indistinguishable to the published review articles with a few up-to-date contents. There are quite a few review articles regarding the nanomaterials and kidney-targeted drug delivery systems, e.g., https://www.frontiersin.org/articles/10.3389/fbioe.2021.683247/full , https://pubs.acs.org/doi/10.1021/acs.molpharmaceut.1c00511 , https://pubmed.ncbi.nlm.nih.gov/30136283/, and https://iopscience.iop.org/article/10.1088/2516-1091/ac6e18 .

3. These overview part can be shortened. The content in 4. Limitations of Conventional Renal Drugs can be combined with 3. Renal Pathophysiology. The arrangement of the main content should be reordered to be 7, 6, and 8.

4 The conclusions of this review manuscript are optimal.

5. The references are appropriate and up-to-date.

6. The Figures and Tables are clear and self-explaining. However, Abbreviation in the Figure, e.g., Fig 3, should be decribed in the Figure caption.

Author Response

Reviewer -1

Dear Prof/Dr.

Thank you for giving us the opportunity to submit a revised version of our manuscript titled " Advanced Drug Delivery Systems For Renal Disorders” to the journal of Gels. We appreciate the time and effort that you have dedicated to providing your valuable feedback on the manuscript. We are grateful to the reviewers for their insightful comments. We have been able to incorporate changes and suggestions provided by the respected reviewers. We have highlighted the changes within the manuscript, and all page numbers refer to the revised manuscript file with tracked changes. Please find below a point-by-point response to the comments and concerns. Moreover, we look forward to hearing from you in due time regarding our submission and to respond to any further questions and comments you may have.

Response to Reviewer 1 Comments

Major concerns

Generally, this review manuscript is acceptable overall, though the writing style and format are not smooth and not easy to follow. The origin of the topic is not that novel, though acceptable. It seems that the authors expand the introduction/overview parts regarding the renal anatomy and functions and renal pathophysiology/limitations of conventional renal drugs to avoid the similarity to the available published review articles. These introduction/overview parts are too long. Shortening the introduction/overview parts and rearrangement of the heading might improve the manuscript's quality and flow.

No

Comment

Point 1

This review manuscript describes a broad overview of the nanomaterials namely nanoparticles and hydrogels as kidney-targeted drug delivery systems. The content is composed of 4 parts: kidney anatomy and functions, renal pathophysiology and limitations of conventional renal drugs, the main barriers to kidney drug delivery, and delivery sites of kidney-targeted drug delivery systems.

However, the overview parts regarding the renal anatomy and functions, renal pathophysiology and limitations of conventional renal drugs, and the main barriers to kidney drug delivery are very long. These overview part can be shortened. The content of the main objective starts from page no. 7, heading no. 6-8 (6. Nanoparticles Factors for Enhanced Renal Accumulation, 7. Delivery Sites of Renal Drug Delivery Systems, 8 Strategies of Renal Drug Delivery Systems). The arrangement of the main content should be also reordered to be 7, 6, and 8.

Response 1

The manuscript was revised. The subtitles 3. Renal Pathophysiology and 4. Limitations of Conventional Renal Drugs were combined together. Besides, the main content was reordered as suggested to be (Delivery Sites of Renal Drug Delivery Systems  Nanoparticles Factors for Enhanced Renal Accumulation  Strategies of Renal Drug Delivery Systems.

During our revision we focused on summarizing the text to shorten the manuscript as possible.

Point 2

The manuscript should be checked carefully to revise writing mistakes. There are a few spelling mistakes (I believe that this mistake will be corrected and checked during preparing the published version by MDPI).  The name of drug delivery systems should always contain the “s”, e.g., Line 21:  nanomaterial, nanoparticle and hydrogel must be corrected as n nanomaterials, nanoparticles and hydrogels.

Response 2

The entire manuscript was checked for the writing mistakes.

Point 3

The topic is relevant in the field. However, if considering the original or novelty, this review is indistinguishable to the published review articles with a few up-to-date contents. There are quite a few review articles regarding the nanomaterials and kidney-targeted drug delivery systems, e.g.,

https://www.frontiersin.org/articles/10.3389/fbioe.2021.683247/full , https://pubs.acs.org/doi/10.1021/acs.molpharmaceut.1c00511 , https://pubmed.ncbi.nlm.nih.gov/30136283/, and https://iopscience.iop.org/article/10.1088/2516-1091/ac6e18

Response 3

Although all mentioned papers were regarding the nanomaterials for kidney-targeted drug delivery systems, none of them combined the renal pathophysiology along with nanoparticle factors for enhanced renal accumulation concerning the physicochemical properties of the nanoparticles.

https://www.frontiersin.org/articles/10.3389/fbioe.2021.683247/full presented the recent developments of nanoparticles designed for kidney targeted delivery and briefly introduce

targeting strategies for the kidney.

https://pubs.acs.org/doi/10.1021/acs.molpharmaceut.1c00511 provided an overview of the unique structural characteristics and injured cells of acute and chronic injured kidneys. Then reviewed the literature on renal cell-targeted formulations of AKI and CKD. Finally, provide some perspectives for future studies.

https://pubmed.ncbi.nlm.nih.gov/30136283/ provided a broad overview of the targeting vectors and targeting pathways for renal tubules and glomeruli. Finally, it summarized the literature examples of drug delivery to the kidneys and elaborated strategies suitable for renal targeting.

https://iopscience.iop.org/article/10.1088/2516-1091/ac6e18 summarized the current trends and recent advancements made in the

development of carrier materials for kidney disease targeted therapies, for AKD, and CKD, and renal cell carcinoma.

Additionally, it discussed the current limitations in carrier materials and their delivery mechanisms.

Point 4

These overview part can be shortened. The content in 4. Limitations of Conventional Renal Drugs can be combined with 3. Renal Pathophysiology. The arrangement of the main content should be reordered to be 7, 6, and 8.

The conclusions of this review manuscript are optimal

The references are appropriate and up-to-date.

Response 4

The manuscript was revised. The subtitles 3 and 4 were combined and the arrangement of the main content was reordered to be (Delivery Sites of Renal Drug Delivery Systems Nanoparticles Factors for Enhanced Renal Accumulation  Strategies of Renal Drug Delivery Systems.

During our revision we focused on summarizing the text to shorten the manuscript as possible.

Point 5

The Figures and Tables are clear and self-explaining. However, Abbreviation in the Figure, e.g., Fig 3, should be described in the Figure caption.

Response 5

The abbreviations were added to Figure 3. The changes were amended on page 6, line 197.

Reviewer 2 Report

Dear authors, the manuscript "Advanced Drug Delivery Systems For Renal Disorders" covers interesting topic. However, I have several comments and questions:

1. Lines 140-141. "Diabetes (50%) is the most prevalent reason for chronic renal impairment in the US before hypertension (25%)." Please, provide the reference to support this sentence.

2. The info in Table 1 does not correlate with provided references:

- According to your Table 1, there were three types of AuNP with sizes of 2.5, 2.9, and 3.1 nm (ref. 62). However, in the ref. 62 itself there were glutathione-coated AuNP with hydrodynamic diameter 2.1 nm, and AuNPs coated with glutathione and cysteamine (2.3 nm core size, 2.9 nm hydrodynamic diameter). More importantly, there was no investigation on renal accumulation, the authors of ref. 62 studied the interaction of nanoparticles with xenograft mouse models of two prostate cancer types, PC-3 and LNCaP. 

- The size of NP in ref. 74 does not correspond to Table 1 (2.4 nm in the Table, 384 nm in the ref. 74) 

- Be consistent with renal accumulation, it should be either %ID or %ID per gram of tissue. For ref.63 you put %ID per gram, for ref. 85 you put just %ID. %ID per gram in ref. 85 should be 1.9 %ID/g.

- Also be consistent with time point at which accumulation or clearance data had been measured in the references. For instance, choose 24 h, if this time point is present in all of your references.

- There are no CdSe/ZnS particles in ref. 86. There are no AuNP with zeta-potential of -50 mV in ref. 86 either.

- Ref. 87 is about silicon NP, not silica (SiO2) NP.

- Why do you need material density column in Table 1?

I find your Table 1 misleading and poorly made. And as your section 6 is based on Table 1, I find it also doubtful. 

I suggest you to carefully read the articles you cite.

Author Response

Reviewer 2

Dear Prof/Dr.

Thank you for giving us the opportunity to submit a revised version of our manuscript titled " Advanced Drug Delivery Systems For Renal Disorders” to journal of Gels. We appreciate the time and effort that have dedicated to providing your valuable feedback on the manuscript. We are grateful to the reviewers for their insightful comments. We have been able to incorporate changes and suggestions provided by the respected reviewers. We have highlighted the changes within the manuscript and all page numbers refer to the revised manuscript file with tracked changes. Please find below a point-by-point response to the comments and concerns. Moreover, we look forward to hearing from you in due time regarding our submission and to responding to any further questions and comments you may have.

Response to Reviewer 2 Comments

Major concerns

Dear authors, the manuscript "Advanced Drug Delivery Systems For Renal Disorders" covers interesting topic. However, I have several comments and questions:

No

Comment

Point 1

Lines 140-141. "Diabetes (50%) is the most prevalent reason for chronic renal impairment in the US before hypertension (25%)." Please, provide the reference to support this sentence.

Response 1

The doi of the ref is: doi: 10.1038/kisup.2015.2. However, this statement was deleted during the revision and summarizing of the manuscript as was suggested by other reviewers.

Point 2

The info in Table 1 does not correlate with provided references:

- According to your Table 1, there were three types of AuNP with sizes of 2.5, 2.9, and 3.1 nm (ref. 62). However, in the ref. 62 itself there were glutathione-coated AuNP with hydrodynamic diameter 2.1 nm, and AuNPs coated with glutathione and cysteamine (2.3 nm core size, 2.9 nm hydrodynamic diameter). More importantly, there was no investigation on renal accumulation, the authors of ref. 62 studied the interaction of nanoparticles with xenograft mouse models of two prostate cancer types, PC-3 and LNCaP. 

- The size of NP in ref. 74 does not correspond to Table 1 (2.4 nm in the Table, 384 nm in the ref. 74) 

- Be consistent with renal accumulation, it should be either %ID or %ID per gram of tissue. For ref.63 you put %ID per gram, for ref. 85 you put just %ID. %ID per gram in ref. 85 should be 1.9 %ID/g.

- Also be consistent with time point at which accumulation or clearance data had been measured in the references. For instance, choose 24 h, if this time point is present in all of your references.

- There are no CdSe/ZnS particles in ref. 86. There are no AuNP with zeta-potential of -50 mV in ref. 86 either.

- Ref. 87 is about silicon NP, not silica (SiO2) NP.

- Why do you need material density column in Table 1?

I find your Table 1 misleading and poorly made. And as your section 6 is based on Table 1, I find it also doubtful. 

I suggest you to carefully read the articles you cite.

Response 2

Thanks for highlighting this mistake which we really overlooked. We used Endnote for citation, but it looks like that some references were imported incompletely from their source. We have made corrections on the table 1 as suggested and the new table (Table 2) is listed on pages 12 & 13.

All particle sizes/zeta potential values were corrected. Regarding ref 62. There was investigation on renal accumulation and excretion in the supplementary data of the manuscript.

The reason for using two units for renal accumulation, (%ID or %ID per gram of tissue) is that some studies used %ID, measures the nanoparticle quantity in each organ, while others reported (% ID/g) as the unit for biodistribution results of nanoparticles.

Two time points was used for the accumulation or clearance data as some of the studies did not evaluate at 24 h. However, in the new table (Table 2) only two time points were used 24h or 48 h.

Material density column was deleted

Table 1 and section 6 was corrected accordingly.

Round 2

Reviewer 2 Report

The manuscript has been substantially improved. Authors have done all the corrections required. The manuscript can be accepted in the present form